# Transcriptomic and Epigenomic Responses to Cortisol-Mediated Stress in Rainbow Trout (*Oncorhynchus mykiss*) Skeletal Muscle

**DOI:** 10.3390/ijms25147586

**Published:** 2024-07-10

**Authors:** Daniela Aravena-Canales, Valentina Valenzuela-Muñoz, Cristian Gallardo-Escarate, Alfredo Molina, Juan Antonio Valdés

**Affiliations:** 1Laboratorio de Biotecnología Molecular, Facultad de Ciencias de la Vida, Universidad Andres Bello, Santiago 8370035, Chile; daniela.aravena.canales@gmail.com (D.A.-C.); amolina@unab.cl (A.M.); 2Interdisciplinary Center for Aquaculture Research (INCAR), University of Concepción, Concepcion 4030000, Chile; valentina.valenzuela@uss.cl (V.V.-M.); crisgallardo@udec.cl (C.G.-E.); 3Escuela de Medicina Veterinaria, Facultad de Ciencias de la Naturaleza, Universidad San Sebastián, Concepción 4030000, Chile; 4Laboratory of Biotechnology and Aquatic Genomics, Department of Oceanography, University of Concepción, Concepcion 4030000, Chile; 5Centro de Investigación Marina Quintay (CIMARQ), Universidad Andres Bello, Quintay 2340000, Chile

**Keywords:** cortisol, *Oncorhynchus mykiss*, skeletal muscle, RNA-seq, WGBS

## Abstract

The production and release of cortisol during stress responses are key regulators of growth in teleosts. Understanding the molecular responses to cortisol is crucial for the sustainable farming of rainbow trout (*Oncorhynchus mykiss*) and other salmonid species. While several studies have explored the genomic and non-genomic impacts of cortisol on fish growth and skeletal muscle development, the long-term effects driven by epigenetic mechanisms, such as cortisol-induced DNA methylation, remain unexplored. In this study, we analyzed the transcriptome and genome-wide DNA methylation in the skeletal muscle of rainbow trout seven days after cortisol administration. We identified 550 differentially expressed genes (DEGs) by RNA-seq and 9059 differentially methylated genes (DMGs) via whole-genome bisulfite sequencing (WGBS) analysis. KEGG enrichment analysis showed that cortisol modulates the differential expression of genes associated with nucleotide metabolism, ECM-receptor interaction, and the regulation of actin cytoskeleton pathways. Similarly, cortisol induced the differential methylation of genes associated with focal adhesion, adrenergic signaling in cardiomyocytes, and Wnt signaling. Through integrative analyses, we determined that 126 genes showed a negative correlation between up-regulated expression and down-regulated methylation. KEGG enrichment analysis of these genes indicated participation in ECM-receptor interaction, regulation of actin cytoskeleton, and focal adhesion. Using RT-qPCR, we confirmed the differential expression of *lamb3*, *itga6*, *limk2*, *itgb4*, *capn2*, and *thbs1*. This study revealed for the first time the molecular responses of skeletal muscle to cortisol at the transcriptomic and whole-genome DNA methylation levels in rainbow trout.

## 1. Introduction

Aquaculture, the farming of aquatic organisms, has emerged as a critical component of global food production, addressing the increasing demand for seafood while significantly contributing to economic growth and food security [1]. However, the intensive nature of aquaculture operations exposes aquatic organisms to various stressors, leading to physiological and behavioral responses that can impact their health, welfare, and productivity [2,3]. Cortisol is one of the most significant hormones involved in fish’s adaptive response to stress, helping to mobilize energy resources and promote survival during challenging situations [4,5]. The most common stressors in teleost farming include chemical, biological, and physical stressors [6]. Numerous studies have analyzed the relationship between the type, magnitude, and duration of the stressor and cortisol levels, demonstrating a close relationship. For example, in rainbow trout (*O. mykiss*), it was determined that temperature stress induced an increase in plasma cortisol levels to 110 ng/mL [7]. Similarly, in the marine teleost red cusk-eel (*Genypterus chilensis*), handling stress induced an increase in plasma cortisol levels to 165 ng/mL [8]. In the flatfish fine flounder (*Paralichthys adspersus*), confinement stress increased plasma cortisol levels to 260 ng/mL [9]. Cortisol is a steroid hormone produced by the inter-renal cells in the fish head kidney in response to stressful situations. It plays a crucial role in various physiological processes, including metabolism, immune response, and growth [10]. The cortisol-mediated stress response is pivotal in regulating both aerobic and anaerobic metabolism, promoting gluconeogenesis, and suppressing glycogen synthesis [11,12]. Additionally, cortisol stimulates glycogenolysis, which is the breakdown of glycogen into glucose, releasing it into the bloodstream [13].

In fish skeletal muscle, cortisol can have both positive and negative effects, depending on the circumstances and duration of its release [14]. Prolonged cortisol release can result in negative consequences such as muscle breakdown, leading to the release of amino acids that can be utilized for energy production or the synthesis of new proteins in other tissues [15]. Cortisol also inhibits protein synthesis in muscle cells, potentially contributing to muscle wasting under conditions of prolonged stress or elevated cortisol levels [16]. In this context, cortisol promotes catabolic-related signaling pathways, such as the ubiquitin–proteasome (UPS) and autophagy–lysosomal (ALS) systems, triggering skeletal muscle atrophy [16]. Although it is well documented that cortisol promotes catabolic processes in fish skeletal muscle, the molecular mechanisms underlying these events and the other relevant processes related to physiological adaptation to stress have recently been described [17].

Cortisol has a wide variety of effects through two distinct mechanisms known as the genomic and nongenomic pathways. The genomic pathway, which has been widely studied, involves the participation of the intracellular corticosteroid receptors in the regulation of gene expression [18]. The nongenomic pathway is a rapid mechanism modulated by the activation of membrane-mediated signaling [19]. Nevertheless, an epigenomic response mediated by stress has been identified in different fish species, suggesting a more complex mechanism of adaptation [20]. DNA methylation is an epigenetic modification that involves the addition of a methyl group to the DNA molecule. Epigenetic modifications can influence gene expression without altering the underlying DNA sequence [21]. Stress-induced DNA methylation changes can have long-lasting effects, potentially contributing to the development of stress-related conditions. To date, there are few studies that address the effects of stress on salmonids’ skeletal muscle and its relationship with methylation and differential expression of genes. A study in Atlantic Salmon (*Salmo salar*) determined the effects of cold-shock and air-exposure stressors on the transcriptome and methylome in skeletal muscle and liver tissues [22]. Another research group examined the effects of acute stress during embryogenesis and chronic stress during the larval stage on global gene expression and DNA methylation in the gills of *S. salar* [23]. However, there are no studies that analyze the direct effects of cortisol as an epigenetic regulator in fish. In the present work, we investigated the molecular responses in skeletal muscle one week after the intraperitoneal administration of cortisol in rainbow trout using RNA-Seq and WGBS analysis. Multiple differentially expressed and methylated genes were identified, revealing a role in the expression of genes associated with ECM-receptor interaction; the regulation of actin cytoskeleton; and focal adhesion, all critical processes for maintaining muscle structure, function, and the response to physiological changes.

## 2. Results

### 2.1. Assessment of Physiological and Transcriptomic Response to Stress

To better understand the molecular effects of cortisol in DNA methylation and gene expression in rainbow trout skeletal muscle, juvenile fish were stimulated with a single dose of cortisol at physiological concentrations. Three hours after treatment, plasma cortisol levels significantly increased in the cortisol-injected groups compared to the control (Figure 1a). Similarly, glucose (Figure 1b) and lactate (Figure 1c) plasma levels also significantly increased in the cortisol groups compared to the control. As expected, seven days after treatment, cortisol (Figure 1a), glucose (Figure 1b), and lactate (Figure 1c) returned to basal levels, showing no significant differences compared to the control group. To determine the transcriptional response associated with cortisol administration, we sequenced the RNA from the skeletal muscle of rainbow trout with samples obtained seven days after treatment. A total of 329,831,740 reads were obtained, corresponding to 6 cDNA libraries. The raw data are available at NCBI under BioProject code PRJNA985763 (BioSamples accessions SAMN41775533, SAMN41775534, SAMN41775535, SAMN41775536, SAMN41775537, and SAMN41775538). After discarding adapters and low-quality reads, we obtained 329,728,653 clean reads for RNA-seq analysis. The average mapping rate was 82.03% in the rainbow trout genome and a GC content of 48.23% (Table 1).

We identified 550 differentially expressed genes (DEGs) between cortisol and control groups, including 505 up-regulated genes and 45 down-regulated genes (Appendix A). DEGs were used to perform a KEGG and GO enrichment analysis in response to cortisol. KEGG pathways analysis revealed that nucleotide metabolism, ECM-receptor interaction, and the regulation of the actin cytoskeleton were highly represented by cortisol action in rainbow trout skeletal muscle (Figure 2).

The most represented GO biological process, cellular component, and molecular function were wound-healing, intermediate filament, and structural molecule activity, respectively (Table 2).

### 2.2. Analysis of DNA Methylation Induced by Cortisol

To analyze the effect of cortisol in the DNA methylation of skeletal muscle rainbow trout, bisulfite-treated DNA genomic libraries were obtained using samples of control and cortisol-treated groups obtained seven days after treatment. A total of 463,260,600 reads were obtained, corresponding to 6 WGBS libraries. After discarding low-quality reads, we obtained 460,837,656 clean reads for Methyl-seq analysis. The average read alignment rate was 75.78% for the rainbow trout genome, and the bisulfate conversion was over 99% for all libraries (Table 3).

Methylated cytosine in the six libraries was CG, CHG, and CHH, where H represents A, T, or C. The average methylation level of CG, CHG, and CHH were 82.32%, 10.84%, and 6.85%, respectively (Figure 3a). Considering that methylation in GC sites is present in a greater proportion, the analyses of differential methylation and its relationship to gene expression were carried out based on the context of GC methylation. Methylation analysis revealed that 79,605 DMRs were present in the 32 annotated chromosomes (Appendix A). Chromosome 2 has the highest number of methylated sites, with 3809 DMRs, and chromosome 32 has the lowest number of methylated sites, with 1535 DMRs. Compared with the control group, 37,751 DMRs showed higher methylation levels, and 41,854 DMRs showed lower methylation levels in the cortisol group (Figure 3b). A total of 2026 DMRs were found in body genes, 260 and 1766 DRMs in exons and introns, respectively. A total of 258 and 370 DMRs were found, 2K Upstream and Downstream, respectively (Figure 3c).

Through DMR annotation, we identified a total of 9059 differentially methylated genes (DMGs), including 2886 differentially methylated promoters (DMPs). KEGG enrichment analysis of 9059 DMGs revealed that focal adhesion, adrenergic signaling in cardiomyocytes, and Wnt signaling pathway were highly represented by cortisol action in rainbow trout skeletal muscle (Figure 4).

Complementary GO enrichment analysis revealed that protein phosphorylation, plasma membrane, and protein binding were over-represented in biological processes, cellular components, and molecular functions, respectively (Table 4). 

### 2.3. Correlation and Validation between DNA Methylation and Gene Expression

Integrative analysis of RNA-seq and DNA-methylation indicated an overlap of 126 genes between 505 up-regulated genes and 5344 down-methylated genes. The KEGG pathways analysis revealed that ECM-receptor interaction, arachidonic acid metabolism, and regulation of actin cytoskeleton were highly represented by cortisol action in rainbow trout skeletal muscle (Table 5). Other interesting pathways over-represented were focal adhesion and nucleotide metabolism.

GO enrichment analysis of these genes revealed that keratinization, intermediate filament organization, and IRE1-mediated unfolded protein response were over-represented in the biological process (Table 6). In cellular components and molecular functions, keratin filament and structural constituent of the epidermis (Table 6) were significantly enriched, respectively. 

Considering that focal adhesion, ECM-receptor interaction, and the regulation of actin cytoskeleton were over-represented in all KEGG enrichment analyses, we selected six candidate genes for RT-qPCR validation. We selected *lamb3* (laminin subunit beta 3) and *itga6* (integrin alpha 6), both involved in ECM-receptor interaction. *Limk2* (LIM domain kinase 2) and *itgb4* (Integrin Subunit beta 4) are involved in the regulation of the actin cytoskeleton. *Capn2* (calpain-2 catalytic subunit) and *thbs1* (thrombospondin-1) are involved in focal adhesion. Our results reveal a high correlation (r = 0.9156) between the expression values of candidate genes using both RNA-seq and qPCR techniques (Figure 5).

## 3. Discussion

The present study reveals the impact of cortisol on the global transcriptomic response of fish skeletal muscle and the epigenetic mechanisms that regulate this expression. Our results show that intraperitoneal administration of cortisol at a concentration of 10 mg/kg achieves plasma levels of the hormone similar to those observed under acute stress conditions [7,24]. In a complementary manner, we also determined an increase in plasma glucose and lactate levels, similar to what has been described in other studies involving acute stressors in various aquatic species [7,24]. This methodology to induce a cortisol-mediated acute stress condition has been supported by various studies. For example, a recent study in marine euryhaline milkfish (*Chanos chanos*) analyzed changes in the expression of *ostf1* (osmotic stress transcription factor 1) in gills after the intraperitoneal administration of 4 μg of cortisol/g of body weight [25]. Similarly, to analyze the chronic effects of this hormone, it has been used with cortisol dissolved in coconut oil and implanted intraperitoneally. In Gilthead Seabream (*Sparus aurata*), the long-term metabolic and transcriptional effects were analyzed through the administration of 40 μg of cortisol/g of body weight dissolved in coconut oil [26]. These delivery systems have allowed the increase in plasma cortisol levels in ranges similar to physiological stress.

As expected, cortisol, glucose, and lactate levels returned to basal values seven days after administration, reflecting the rapid catabolism of steroid hormones and the transient metabolic response to stress [27]. Interestingly, seven days after cortisol administration, we identified changes in the expression of half a thousand genes, the majority of which were overexpressed. These genes were mainly associated with biological processes such as wound-healing, intermediate filament cytoskeleton organization, keratinization, intermediate filament organization, and cell–cell junction assembly, all of which are crucial for maintaining muscle structure and function. Recently, our research group determined early transcriptomic and proteomic changes modulated by cortisol at 3 and 9 h after hormone administration [28,29,30]. Although the number of differentially expressed genes varied, the ontological enrichment analysis revealed that some biological processes, such as cell–cell adhesion and focal adhesion, were common in both GO and KEGG analyses. Interestingly, processes associated with protein catabolism, such as protein ubiquitination or ubiquitination-dependent protein catabolism, were not represented seven days after cortisol treatment. These observations are consistent with in vitro assays in rainbow trout myotubes, where it was determined that cortisol, in an early manner, can modulate the transient activation of signaling pathways related to the production of reactive oxygen species (ROS) through non-genomic mechanisms [31], as well as the expression of the atrogenes *atrogin-1*/*murf1* [17], and *pdk2*, a key regulator of energy metabolism [32]. It is possible to conjecture that cortisol associated with an acute stress event can modulate the expression of genes associated with protein catabolism in the early stages through non-genomic and genomic mechanisms, and in the intermediate stages, it modulates the expression of genes associated with maintaining muscle structure and function through epigenetic mechanisms.

To demonstrate this point, we examined the dynamics of DNA methylation as a potential epigenetic mechanism in response to cortisol-mediated stress. We conducted a WGBS analysis of control and stressed groups at 7 days. The methylation rates of CG were much higher than those of CHG and CHH, consistent with previous reports in salmonid and other teleost species [33,34]. These methylations were homogeneously distributed across all chromosomes, similar to other studies [35,36]. We also determined that differential methylations were mainly distributed in the gene body, although a significant number were found in promoter regions close to the transcription start site. To determine the role of DNA methylation in the biological processes associated with cortisol treatment, we performed a correlation analysis between differentially expressed genes (DEGs) and differentially methylated genes (DMGs) in their promoter regions or gene bodies. We found that 126 genes met these characteristics, mainly associated with ECM-receptor interaction, regulation of the actin cytoskeleton, and focal adhesion. These pathways are critical for maintaining cellular integrity and function during stress, indicating that cortisol orchestrates a complex network of genetic responses to promote adaptation and survival under stress conditions. Our results align with studies on Atlantic Salmon (*S. salar*) under cold-shock and air-exposure stressors, revealing positive growth changes mediated by epigenetic components linked to hormesis, an adaptive response to low-level repeated stress [22]. In later work, the same group described a dynamic response between the expression and methylation of genes associated with growth (GH, IGF-1) and genes involved in the immune response (Toll receptors, a major histocompatibility class), validating the relationship between mild stress and enhanced growth performance [37]. Similarly, results obtained in the gills of *S. salar* demonstrated through an epigenomic and transcriptomic approach that chronic stress induced changes in the promoter and gene-body methylation of immune-related genes, suggesting that stress can affect immune competence through epigenetic mechanisms [23]. Interestingly, our group recently described the environmental influence on the epigenomic and transcriptomic response during sea lice infection (*Caligus rogercresseyi*) of Atlantic Salmon (*S. salar*) [38]. All these studies show a strong relationship between the epigenome, transcriptome, and type of stress. Although there is no clear understanding of the mechanisms by which cortisol modulates gene methylation to date, there are precedents from studies in mammals. Glucocorticoids (GCs) can induce DNA (de)methylation changes affecting gene expression. DNA methylation involves DNA methyltransferases (DNMTs), adding methyl groups to cytosine, forming 5-methylcytosine (5mC), which suppresses gene expression. Conversely, ten-eleven translocation (TET) proteins convert 5mC to 5-hydroxymethylcytosine (5hmC), leading to unmethylated cytosine and enhancing gene expression [39]. The active DNA demethylation mechanism is associated with the up-regulation of TET enzyme expression following GC treatment, as reported in retinal pigment epithelial (RPE) cells and human osteocytes [40,41]. The passive demethylation event involves GC-induced hypomethylation through down-regulation of DNMT1 expression, described in a pituitary adenoma cell line [42]. Further research is needed to clarify the mechanism of gene methylation in teleosts mediated by cortisol.

To validate our results, we selected six genes that showed a negative correlation in their methylation levels and gene expression, and related to focal adhesion, ECM-receptor interaction, and regulation of the actin cytoskeleton. Focal adhesions are specialized cell-substrate contacts formed by integrin proteins and actin filaments that modulate cellular effects in response to extracellular matrix adhesion [43]. Interestingly, previous studies have shown a close relation between the action of steroid hormones and the generation of focal adhesion in the muscles of mammals and teleost [44,45]. To date, various focal adhesion-related genes have been identified as responsive to glucocorticoids, indicating a crucial role for this process in the adaptive stress response in mammals’ skeletal muscle [46,47]. However, the impact of cortisol on focal adhesion in teleost skeletal muscle is poorly understood. Among the genes with differential expression validated by RT-qPCR are *lamb3*, *itga6*, *limk2*, *itgb4*, *capn2*, and *thbs1*. *Lamb3*, also known as laminin subunit beta 3, encodes for the subunit beta of a protein called laminin 5 [48]. This protein is fundamental for the formation of intramuscular connective tissue networks in teleost [49]. In teleost, its expression has been related to follicular pseudoplacenta development in Black Rockfish (*Sebastes schlegelii*) [50]; however, there are no studies that relate its expression or gene methylation to cortisol-mediated stress response. *Itga6* and *itgb4*, also known as integrin Subunit Alpha 6 and integrin Subunit beta 4, respectively, encode for members of the integrin family of proteins [51]. In mammals, their expression is important for myogenic stem cell differentiation [52]. In particular, *Itga6* expression has been related to focal adhesion involved in the immune response of *Cynoglossus semilaevis* to the infection of *Vibrio vulnificus* [53]. *Limk2*, also known as LIM Domain Kinase 2, is a Serine/threonine-protein kinase that plays an essential role in the regulation of actin filament formation [54]. *Limk2* presence is essential during zebrafish (*Danio rerio*) embryogenesis [55]. *Capn2* encodes a calcium-regulated protease, which catalyzes the proteolysis of substrates involved in cytoskeletal remodeling [56]. The differential expression of this gene has been described during myogenesis in gilthead sea bream (*Sparus aurata*) myoblast [57]. *Thbs1*, or Thrombospondin 1, is a subunit of homotrimeric protein, involved as a glycoprotein that mediates cell-to-cell and cell-to-matrix interactions [58]. Interestingly, this gene has been associated with growth in rainbow trout through genome-wide identification tools [59]. To our knowledge, there are no previous reports linking the expression or methylation of these genes with cortisol-mediated stress in teleost. This study shed light on the complex interplay between gene expression and DNA methylation in response to cortisol in rainbow trout skeletal muscle. The findings highlight the dual role of cortisol in modulating transcriptional and epigenetic landscapes. We propose that cortisol modulates the expression of genes associated with maintaining muscle structure and function through epigenetic mechanisms.

## 4. Materials and Methods

### 4.1. Ethics Statement

This study adhered to animal welfare procedures and was approved by the bioethical committees of the Universidad Andres Bello and the National Commission for Scientific and Technological Research of the Chilean government (protocol code 010/2023).

### 4.2. Protocol Experiment

Juvenile rainbow trout (*O. mykiss*) (21.5 g ± 3.2) were obtained from Pisciculture Rio Claro (IX region, Chile). Fish were maintained at physiological temperatures (14 °C ± 1 °C) and photoperiod (Light:Dark 12:12). All individuals were fed daily with commercial pellets (Skretting Chile, Osorno, Chile) during the duration of the trial, except for the day before the in vivo protocol, and divided into two tanks (control group and cortisol group). In control groups (n = 20), fish were treated with a vehicle solution (DMSO and PBS 1X), and in cortisol groups (n = 20), fish were treated with 10 mg/kg of cortisol (Sigma-Aldrich, St. Louis, MO, USA). This dose has been previously validated and allows obtaining plasma cortisol ranges similar to those observed under conditions of physiological stress. After three hours of cortisol administration, 10 fish from control group and 10 fish from cortisol group were randomly selected and euthanized by overdoses of benzocaine (300 mg/L). Similarly, seven days after cortisol administration, the rest of the individuals in control (n = 10) and cortisol groups (n = 10) were euthanized. In both tests, 1 mL of blood was collected with heparin (10 mg/mL), and plasma was separated from all groups via centrifugation at 5000× *g* for 10 min and stored at −80 °C. Epaxial skeletal muscle was obtained from sampled fish and frozen in liquid nitrogen. Plasma cortisol, glucose, and lactate were measured using the Cayman cortisol (Cayman Chemical, Ann Arbor, MI, USA), glucose (Abcam, Cambridge, UK), and L-Lactate Assay Kits, respectively (Abcam, Cambridge, UK), as previously described [8]. 

### 4.3. RNA Extraction, Library Construction, Sequencing and RNA-Seq Analysis

RNA was extracted from the skeletal muscle of the control and cortisol groups using the RNeasy Mini Kit (Qiagen, Germantown, MD, USA), following the manufacturer’s recommendations. RNA integrity was confirmed by electrophoresis in agarose gel (1.2%) and the capillary electrophoresis Fragment Analyzer Automated CE System (Advanced Analytical Technologies, Inc., Ames, IA, USA). Only RNA samples with RQN ≥ 8.5 were used in further analyses. The amount of total RNA was measured by a fluorometer using the Qubit RNA BR assay kit (Invitrogen, Carlsbad, CA, USA). cDNA libraries were generated using the TruSeq RNA Sample Preparation kit v2 (Illumina, San Diego, CA, USA). A total of 6 libraries were sequenced using a paired-end strategy (2 × 100 bp) with the HiSeq 2500 (Illumina) platform of Macrogen (Seoul, Republic of Korea). Raw reads were trimmed using CLC genomic workbench software v23 (CLC bio—Qiagen, Germantown, MD, USA) (Q > 20 and read length > 50 bp). Trimmed reads were mapped onto the reference rainbow trout genome USDA_OmykA_1.1 (RefSeq GCF_013265735.2) using default mapping parameters: mismatches = 2, minimum fraction length = 0.9, minimum fraction similarity = 0.8, and maximum hits per read = 5. Differential expression analysis in silico was based on reads that uniquely mapped to the reference and the proportional-based statistical K-test. Transcripts with absolute fold-change values *≥* 2.0 and an FDR-corrected *p*-value *<* 0.05 were considered differentially expressed in silico in the analysis. The ID of differentially expressed genes was extracted and used as input to the DAVID GO and KEGG enrichment analysis (https://david.ncifcrf.gov/, accessed on 18 October 2023). 

### 4.4. Real-Time qPCR Validation

The total RNA extracted was described above. The RNA was quantified using NanoDrop technology (BioTek Instruments, Winooski, VT, USA), selecting the samples with A260/280 ratio between 1.9 and 2.1. For cDNA synthesis, 1 μg of RNA was reverse-transcribed into cDNA for 60 min at 42 °C using the ImProm-II Reverse Transcription System (Promega, Madison, WI, USA) under manufacturer recommendations. The real-time PCR (qPCR) was performed in an MX3000P thermocycler (Agilent Technologies, Santa Clara, CA, USA) following the MIQE guidelines [60], using Brilliant^®^ II SYBR^®^ master mix (Agilent Technologies) and qPCR protocol previously described previously [61]. Primer sequences are listed in Appendix A. A melting curve was obtained to confirm a single PCR product. Data were expressed in arbitrary units (AU) and analyzed using Q-Gene software v4.4.0 [62]. Beta actin (actβ) and 40S ribosomal protein S30 (fau) were used as housekeeping genes.

### 4.5. DNA Extraction, Library Construction, Ssequencing, and WGBS Analysis

Genomic DNA was extracted using DNeasy Blood & Tissue Kit (Qiagen, Germantown, MD, USA), following the manufacturer’s recommendations. The amount of total DNA was measured by a fluorometer using the Qubit dsDNA BR Assay (Invitrogen, Orange, CA, USA). Six DNA libraries were prepared by Zymo-Seq WGBS Library Kit (ZymoResearch, Orange, CA, USA), using 100 ng of input DNA. After PCR amplification, sequencing was performed using a paired-end strategy (2 × 150 bp) with the Novaseq (Illumina) platform of Macrogen (Seoul, Republic of Korea). Raw reads were trimmed using CLC genomic workbench software v23 (CLC bio—Qiagen, Germantown, MD, USA) (Q > 20 and read length > 50 bp). Trimmed reads were analyzed using the Bisulfite sequencing tools on CLC genomics Workbench v23 (Qiagen Bioinformatics, Redwood City, CA, USA). Bisulfite data were mapped to rainbow trout genome USDA_OmykA_1.1 (RefSeq GCF_013265735.2) to locate the CpG markers. The methylation levels were determined with the tool Call Methylation levels of CLC Genomics Workbench v23 (CLC bio—Qiagen, Germantown, MD, USA). Finally, the differentially methylated nearby genes were extracted from the *O. mykiss* genome using the CLC Genomics Workbench v23 (CLC bio—Qiagen, Germantown, MD, USA). The ID of differentially methylated genes was extracted and used as input to the DAVID GO and KEGG enrichment analysis (https://david.ncifcrf.gov/, accessed on 18 October 2023).

### 4.6. Statistical Analysis

All data were analyzed using one-way ANOVA and a Tukey’s honestly significant difference (HSD) as a post hoc test, using Graph Prism 7.0 software (GraphPad Software, Inc., San Diego, CA, USA). A probability level with a *p*-value *<* 0.05 was used as the minimum to indicate the statistical significance.

## 5. Conclusions

This study elucidates the complex interplay between gene expression and DNA methylation in response to cortisol in rainbow trout skeletal muscle. The findings highlight the dual role of cortisol in modulating transcriptional and epigenetic landscapes, providing a comprehensive understanding of the molecular mechanisms underlying stress responses in fish. We propose that cortisol modulates the expression of genes associated with maintaining muscle structure and function through epigenetic mechanisms. These insights have significant implications for improving aquaculture practices, ensuring the health and productivity of farmed fish. Future research should focus on the long-term effects of cortisol and explore potential epigenetic markers for stress resilience, contributing to the development of robust and sustainable aquaculture systems.

## Figures and Tables

**Figure 1 ijms-25-07586-f001:**
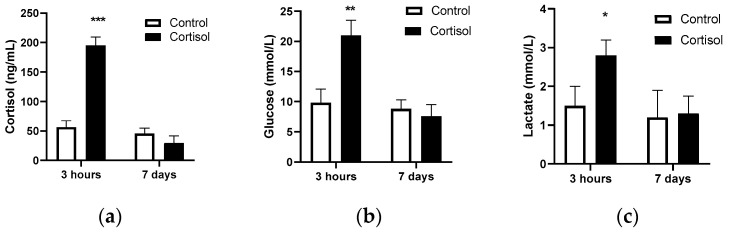
Assessment of physiological response to stress: cortisol (**a**), glucose (**b**), and lactate (**c**) in plasma were assessed in juvenile rainbow trout kept under cortisol and control 3 h and 7 days after treatment. The results are expressed as means and + standard errors (n = 5 per treatment). Differences between control and stress groups are shown in * *p* < 0.05, ** *p* < 0.01, *** *p* < 0.005.

**Figure 2 ijms-25-07586-f002:**
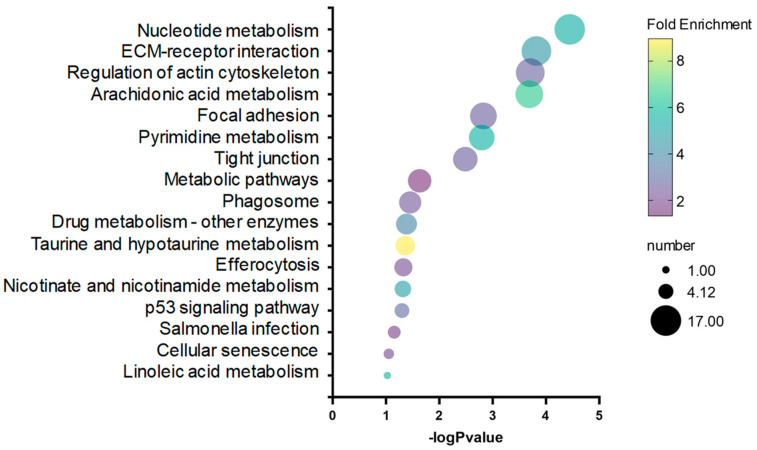
Bubble plot showing enriched KEGG pathways, resulting from the DEGs between control and cortisol groups.

**Figure 3 ijms-25-07586-f003:**
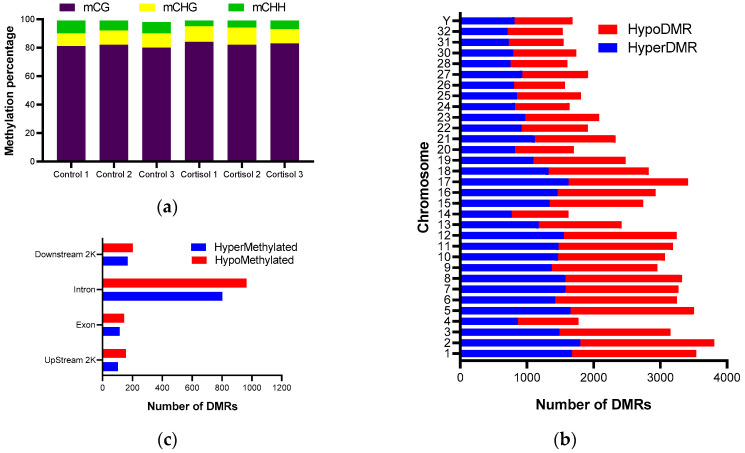
Genome profile of CpG methylation. (**a**) Relative level of methylated cytosines (CG, CHG, and CHH) in experimental groups. (**b**) Chromosome distribution of hyper- and hypo-differentially methylated regions. (**c**) Number of DMRs in different genomics contexts.

**Figure 4 ijms-25-07586-f004:**
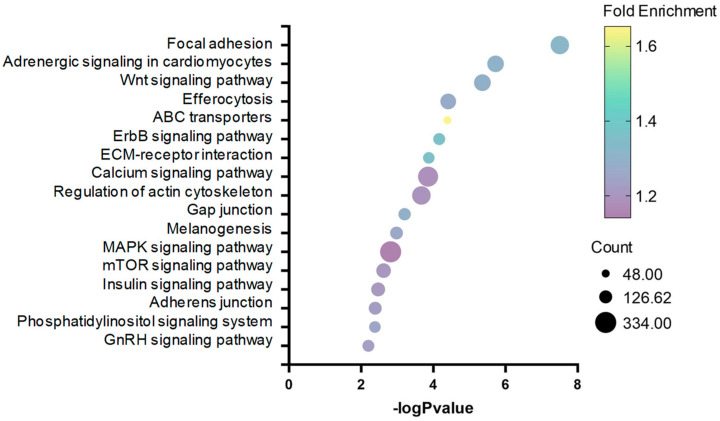
Bubble plot showing enriched KEGG pathways resulting from the DMGs between control and cortisol groups.

**Figure 5 ijms-25-07586-f005:**
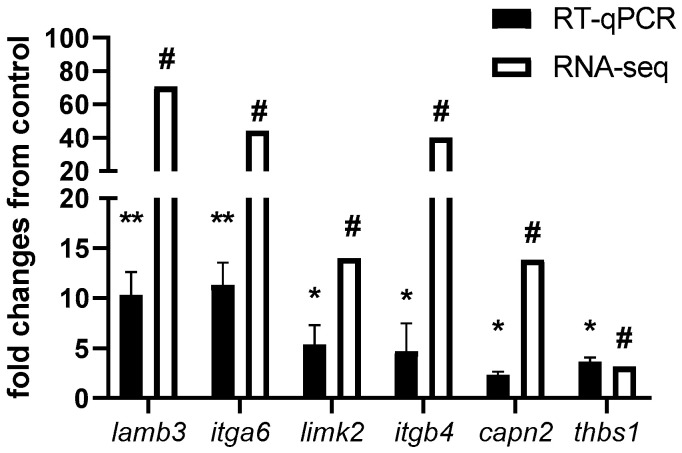
RT-qPCR validation of DEGs and DMGs between control and cortisol treatments. Genes selected for the RT-PCR validation were *lamb3*, *itga6*, *limk2*, *itgb4*, *capn2*, and *thbs1*. For RNA-seq, in White, “#” indicates a log2 fold change ≥2.0 and FDR <0.05. For RT-qPCR, in black, relative expression was normalized against *fau* and *actβ*. The results are expressed as means and + standard errors (n = 5 per treatment). Differences between control and cortisol groups are shown in * *p* < 0.05 and ** *p* < 0.01.

**Table 1 ijms-25-07586-t001:** Summary of the RNA-seq Data.

Name	Number of Reads	Avg. Length	Number of Reads after Trimming	Avg. Length after Trimming	GC Content %	Mapping Rate
Control 1	57,044,114	101	57,027,988	95.98	48.59	82.35
Control 2	50,668,370	101	50,653,122	95.42	47.92	86.46
Control 3	54,686,284	101	54,664,282	94.58	48.15	80.34
Cortisol 1	54,262,334	101	54,249,443	96.33	48.76	82.26
Cortisol 2	56,252,594	101	56,234,957	95.63	47.96	80.49
Cortisol 3	56,918,044	101	56,898,861	95.21	48.02	80.25
Total	329,831,740	101	329,728,653	95.53	48.23	82.03

**Table 2 ijms-25-07586-t002:** Enrichment of DEGs associated with biological process, cellular components, and molecular function between control and cortisol groups.

Category	Go Term	Gene Number	*p*-Value
Biological Process	Wound-healing	8	1.95 × 10^−9^
	Intermediate filament cytoskeleton organization	10	1.12 × 10^−8^
	Keratinization	6	3.28 × 10^−8^
	Intermediate filament organization	7	4.56 × 10^−8^
	Cell–cell junction assembly	10	3.21 × 10^−7^
Cellular Component	Intermediate filament	22	6.42 × 10^−17^
	Extracellular space	46	1.08 × 10^−10^
	Keratin filament	8	5.57 × 10^−8^
	Actin cytoskeleton	14	1.03 × 10^−7^
	Cytoplasm	101	1.04 × 10^−6^
Molecular Function	Structural molecule activity	21	7.27 × 10^−12^
	Cadherin-binding	15	9.10 × 10^−11^
	Structural constituent of epidermis	6	2.30 × 10^−8^
	Calcium-dependent cysteine-type endopeptidase activity	8	3.23 × 10^−6^
	Calcium ion binding	40	4.77 × 10^−6^

**Table 3 ijms-25-07586-t003:** Summary of the WGBS Data.

Name	Number of Reads	Avg. Length	Number of Reads after Trimming	Avg. Length after Trimming	Mapping Rate	BS Conversion Rate (%)
Control 1	76,907,896	151	76,445,383	114.73	76.17	99.91
Control 2	76,721,140	151	76,304,825	114.83	75.24	99.92
Control 3	77,201,332	151	76,843,936	115.51	74.31	99.91
Cortisol 1	77,549,376	151	77,172,273	119.23	77.16	99.93
Cortisol 2	77,471,992	151	77,090,668	118.88	77.89	99.89
Cortisol 3	77,408,864	151	76,980,571	114.97	73.88	99.95
Total	463,260,600	151	460,837,656	116.36	75.78	99.92

**Table 4 ijms-25-07586-t004:** Enrichment of DMGs associated with biological processes, cellular components, and molecular functions between control and cortisol groups.

Category	Go Term	Gene Number	*p*-Value
Biological Process	Protein phosphorylation	464	5.07 × 10^−41^
	Axon guidance	164	8.38 × 10^−19^
	Transmembrane transport	247	2.33 × 10^−12^
	Multicellular organism development	157	3.24 × 10^−12^
	Cell migration	100	7.74 × 10^−12^
Cellular Component	Plasma membrane	1748	9.82 × 10^−56^
	Cytoplasm	2167	1.11 × 10^−44^
	Neuron projection	209	1.01 × 10^−22^
	Cytosol	600	1.82 × 10^−22^
	Adherens junction	123	2.04 × 10^−14^
Molecular Function	Protein binding	1991	2.26 × 10^−13^
	RNA polymerase II transcription factor activity, DNA binding	732	8.20 × 10^−13^
	Transmitter-gated ion channel activity involved in regulation of P.M.P	51	1.95 × 10^−11^
	mRNA binding	159	3.37 × 10^−11^
	RNA polymerase II core promoter proximal region, DNA binding	738	1.10 × 10^−10^

**Table 5 ijms-25-07586-t005:** KEGG Enrichment analysis of 126 DMGs/DEGs between control and cortisol groups.

KEGG Pathway	Number	*p*-Value	Genes
ECM-receptor interaction	5	2.57 × 10^−3^	*lamb3*, *itga6*, *thbs1*, *itgb4*, *itgb3*
Arachidonic acid metabolism	4	3.02 × 10^−3^	*pla2g4f*, *ptgs2*, *cyp2j2*, *ggt5*
Regulation of actin cytoskeleton	7	5.85 × 10^−3^	*limk2*, *gsn*, *itga6*, *arpc1b*, *itgb4*, *c7*, *itgb3*
Focal adhesion	6	1.63 × 10^−2^	*lamb3*, *itga2*, *capn2*, *thbs1*, *itgb4*, *itgb3*
Efferocytosis	5	2.60 × 10^−2^	*mapk12*, *ptgs2*, *thbs*, *itgb3*, *abca1*
GnRH signaling pathway	4	2.99 × 10^−2^	*mapk12*, *ptk2b*, *pla2g4e*, *mmp14*
Ether lipid metabolism	3	3.40 × 10^−2^	*pla2g4e*, *plpp2*, *gdpd3*
VEGF signaling pathway	3	7.61 × 10^−2^	*mapk12*, *pla2g4e*, *ptgs2*
Nucleotide metabolism	3	8.51 × 10^−2^	*gda*, *nt5c1a*, *nt5c2*

**Table 6 ijms-25-07586-t006:** GO enrichment analysis of 126 DMGs/DREGs between control and cortisol groups.

Category	Go Term	Gene Number	*p*-Value
Biological Process	Keratinization	6	3.05 × 10^−11^
	Intermediate filament organization	6	1.70 × 10^−9^
	IRE1-mediated unfolded protein response	3	5.71 × 10^−4^
	Regulation of RNA metabolic process	3	9.25 × 10^−4^
	Intrinsic apoptotic signaling pathway in response to ERS	3	9.25 × 10^−4^
Cellular Component	Keratin filament	6	2.44 × 10^−8^
	Extracellular space	13	3.03 × 10^−4^
	IRE1-TRAF2-ASK1 complex	3	5.15 × 10^−4^
	Plasma membrane	24	7.65 × 10^−4^
	Bicellular tight junction	5	9.33 × 10^−4^
Molecular Function	Structural constituent of epidermis	6	1.90 × 10^−11^
	Calcium ion binding	15	1.07 × 10^−4^
	Cadherin binding	4	4.31 × 10^−3^
	Endoribonuclease activity	3	6.31 × 10^−3^
	Calcium-dependent cysteine-type endopeptidase activity	3	1.03 × 10^−2^

## Data Availability

The raw read sequences obtained from sequencing were deposited in the Sequence Read Archive (SRA) under BioProject accession number PRJNA1122133 (SRR29380880, SRR29380878, SRR29380877, SRR29380881, SRR29380879, SRR29380876, SRR29342690, SRR29342687, SRR29342689, SRR29342688, SRR29342691, and SRR29342686). The datasets generated and analyzed during the current study are publicly available.

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
