# Peer review of "Transcriptomic and Epigenomic Responses to Cortisol-Mediated Stress in Rainbow Trout (Oncorhynchus mykiss) Skeletal Muscle"

_ijms, 2024, doi:10.3390/ijms25147586_

Round 1
Reviewer 1 Report
Comments and Suggestions for Authors
The manuscript (ijms-3088703) investigates the molecular response of rainbow trout (Oncorhynchus mykiss) to cortisol by RNA-Seq and WGBS analysis.
In my opinion, the topic of study is very interesting, and the information provided well-structured. However, there are some points that could be improved. Below my specific comments.
First of all, the manuscript needs linguistic and typo revision:
Line 49: hormones should be hormone
Line 75: need to be rephrased.
Line 282: need to be rephrased and divided into two new sentences
Line 381: maintained “at”
Etc…
Introduction
Before describing the effects of cortisol, the authors should explain what stress studies have been done in fish (indicating the species) and the levels of cortisol detected. They could then link this point to the information provided in the discussion (lines 319-340) where they refer to stress and methylation (i.e. they should transfer some of this information to the introduction, leaving in that section only what is really related to the discussion). Finally, the authors should include the studies in which cortisol was administered in fish, indicating the route of administration and the concentration used.
2. Material and methods
Line 382: Specify the brand of the commercial feed (Skretting, Biomar, etc.)
Discussion
Lines 319-340: Should be mostly transferred to the introduction as previously mentioned.
References
The bibliography should be revised as it is not uniform. There are some references in which the names of some fish species are not in italics (14, 22, 26, 31….)
Comments on the Quality of English LanguageThe manuscript needs linguistic and typo revision
Author Response
We thank the reviewer for the supportive comments and corrections. We believe we have addressed all the reviewer’s concerns, making significant improvements to the manuscript.
First of all, the manuscript needs linguistic and typo revision. R: A complete revision of English language and typo was carried out.
Line 49: hormones should be hormone. R: We appreciate the correction (see line 47)
Line 75: need to be rephrased. R: The sentence was reworded (see lines 79-80)
Line 282: need to be rephrased and divided into two new sentences. R: The sentence was divided into two new sentences. (see lines 310-314)
Line 381: maintained “at”. R: We appreciate the correction (see line 419)
Introduction
Before describing the effects of cortisol, the authors should explain what stress studies have been done in fish (indicating the species) and the levels of cortisol detected. R: We incorporated the requested information (see line 50-56)
They could then link this point to the information provided in the discussion (lines 319-340) where they refer to stress and methylation (i.e. they should transfer some of this information to the introduction, leaving in that section only what is really related to the discussion). R: We moved the requested information from discussion to introduction (see line 86-92)
Finally, the authors should include the studies in which cortisol was administered in fish, indicating the route of administration and the concentration used. R: We incorporated the requested information (see line 298-307)
Material and methods
Line 382: Specify the brand of the commercial feed (Skretting, Biomar, etc.). R: We incorporated the requested information (see line 421)
Discussion
Lines 319-340: Should be mostly transferred to the introduction as previously mentioned. R: We moved the requested information from discussion to introduction (see line 86-92)
References
The bibliography should be revised as it is not uniform. There are some references in which the names of some fish species are not in italics (14, 22, 26, 31….): R: A complete revision of bibliography format was carried out.
Reviewer 2 Report
Comments and Suggestions for Authors
This work investigates the molecular mechanisms by which cortisol affects growth and development in rainbow trout (Oncorhynchus mykiss), focusing on both transcriptomic changes and epigenetic modifications, specifically DNA methylation. Cortisol is known to regulate growth and stress responses in teleosts, but the study aims to fill a gap in understanding how long-term effects are mediated through epigenetic mechanisms.
The authors identified or hypothesized specific pathways and genes that are modulated by cortisol through changes in DNA methylation, contributing new insights into the molecular mechanisms by which cortisol influences skeletal muscle growth and development in teleosts.
Research on the direct link between cortisol and DNA methylation in fish skeletal muscle is scarce. However, studies in mammals suggest a possible connection between chronic stress (elevated cortisol) and DNA methylation patterns in various tissues.
The manuscript is well written but there are few areas which require revision.
These are some suggestion for improving some sections of the manuscrptipt:
1.The introduction could elaborate on the current knowledge gap for both mammals and teleost fish (see lines 437-450), this would be a great way to introduce the reader to the subject and highlight the originality of your research. Furtheremote you can tailor the examples cited to your specific research focus (e.g., genes related to muscle growth or specific types of stress) and biefly mention the potential implications of your research on fish health, aquaculture practices, or human health (if applicable).
2. Overall, the methodology section provides a clear and comprehensive description of the experimentl procedures, you should however briefly explain the rationale behind choosing the specific cortisol dose (10 mg/kg) used in the experiment.
3. The discussion and conclusion should be aligned better with the research questions and the implications of the findings. You could mention the key research questions you aimed to answer in the introduction, summarize the specific findings regarding your hypothesis particularly for gene expression changes and DNA methylation patterns observed after cortisol treatment.
4. Acknowledge any limitations of your study, such as the use of a specific dose or discuss the potential mechanisms by which cortisol might be influencing gene expression through DNA methylation. Did you identify any genes or pathways that could be assessed in future researchs?
Author Response
We thank the reviewer for the supportive comments and corrections. We believe we have addressed all the reviewer’s concerns, making significant improvements to the manuscript.
This work investigates the molecular mechanisms by which cortisol affects growth and development in rainbow trout (Oncorhynchus mykiss), focusing on both transcriptomic changes and epigenetic modifications, specifically DNA methylation. Cortisol is known to regulate growth and stress responses in teleosts, but the study aims to fill a gap in understanding how long-term effects are mediated through epigenetic mechanisms. The authors identified or hypothesized specific pathways and genes that are modulated by cortisol through changes in DNA methylation, contributing new insights into the molecular mechanisms by which cortisol influences skeletal muscle growth and development in teleosts. Research on the direct link between cortisol and DNA methylation in fish skeletal muscle is scarce. However, studies in mammals suggest a possible connection between chronic stress (elevated cortisol) and DNA methylation patterns in various tissues. The manuscript is well written but there are few areas which require revision. These are some suggestion for improving some sections of the manuscript.
1.The introduction could elaborate on the current knowledge gap for both mammals and teleost fish (see lines 437-450), this would be a great way to introduce the reader to the subject and highlight the originality of your research. Furtheremote you can tailor the examples cited to your specific research focus (e.g., genes related to muscle growth or specific types of stress) and biefly mention the potential implications of your research on fish health, aquaculture practices, or human health (if applicable). R: We incorporated the requested information. However, this information was incorporated into the discussion, where the gaps between mammals and teleost are analyzed (see lines 360-373)
- Overall, the methodology section provides a clear and comprehensive description of the experimentl procedures, you should however briefly explain the rationale behind choosing the specific cortisol dose (10 mg/kg) used in the experiment. R: We incorporated the requested information (see lines 424-246)
- The discussion and conclusion should be aligned better with the research questions and the implications of the findings. You could mention the key research questions you aimed to answer in the introduction, summarize the specific findings regarding your hypothesis particularly for gene expression changes and DNA methylation patterns observed after cortisol treatment. R: The discussion and conclusion were aligned with the research questions.
- Acknowledge any limitations of your study, such as the use of a specific dose or discuss the potential mechanisms by which cortisol might be influencing gene expression through DNA methylation. R: We incorporated a phrase where we mentioned the limitations of the study (lines 405 to 410), as well as the potential mechanism by which cortisol might be influencing gene expression through DNA methylation (lines 362 to 371).